# TYK2: An Upstream Kinase of STATs in Cancer

**DOI:** 10.3390/cancers11111728

**Published:** 2019-11-05

**Authors:** Katharina Wöss, Natalija Simonović, Birgit Strobl, Sabine Macho-Maschler, Mathias Müller

**Affiliations:** Institute of Animal Breeding and Genetics, University of Veterinary Medicine Vienna, A-1210 Vienna, Austria; Katharina.Woess@vetmeduni.ac.at (K.W.); natalijabozovic@gmail.com (N.S.); birgit.strobl@vetmeduni.ac.at (B.S.); sabine.macho-maschler@vetmeduni.ac.at (S.M.-M.)

**Keywords:** tyrosine kinase 2, JAK family of protein tyrosine kinases, signal transducer and activator of transcription, cytokine receptor signaling, gain-of-function mutation, tumorigenesis

## Abstract

In this review we concentrate on the recent findings describing the oncogenic potential of the protein tyrosine kinase 2 (TYK2). The overview on the current understanding of TYK2 functions in cytokine responses and carcinogenesis focusses on the activation of the signal transducers and activators of transcription (STAT) 3 and 5. Insight gained from loss-of-function (LOF) gene-modified mice and human patients homozygous for *Tyk2*/*TYK2*-mutated alleles established the central role in immunological and inflammatory responses. For the description of physiological TYK2 structure/function relationships in cytokine signaling and of overarching molecular and pathologic properties in carcinogenesis, we mainly refer to the most recent reviews. Dysregulated TYK2 activation, aberrant TYK2 protein levels, and gain-of-function (GOF) TYK2 mutations are found in various cancers. We discuss the molecular consequences thereof and briefly describe the molecular means to counteract TYK2 activity under (patho-)physiological conditions by cellular effectors and by pharmacological intervention. For the role of TYK2 in tumor immune-surveillance we refer to the recent Special Issue of Cancers “JAK-STAT Signaling Pathway in Cancer”.

## 1. TYK2-Mediated Cytokine Signaling and Activation of STAT3 and STAT5

TYK2 was the first identified member of a family of non-receptor kinases later termed Janus kinases (JAK), which additionally comprises JAK1-3 [1,2]. JAKs are associated with cytokine and growth factor receptors and activate STAT (STAT1-4, STAT5A, STAT5B, STAT6) family members [2,3]. JAKs share four functional domains (from N- to C-terminal): (i) a four-point-one, ezrin, radixin, moesin (FERM) homology domain; (ii) an atypical Src-homology 2 (SH2) domain, both facilitating protein-protein interactions (PPIs); (iii) a kinase-like or pseudokinase (JAK homology (JH) 2) domain negatively regulating the kinase activity; and (iv) a tyrosine kinase (JH1) domain which, upon conformational changes at ligand bound receptors, increases its catalytic activity by trans-/autophosphorylation of its activation loop [2,4].

To date, the requirement for TYK2 in signaling has been shown for numerous cytokines, including distinct interleukin (ILs) and interferons (IFNs), which comprise several subtypes (i.e., type I and III IFNs). The heterodimeric cytokine receptor complexes are composed of four distinct TYK2-associated receptor chains (IFNAR1, IL-12Rβ1, IL-10R2, and IL-13Rα1) and a respective second receptor chain associated either with JAK1 or JAK2, which serves as the signal transducing chain harboring STAT docking sites. Usually, these sites contain critical tyrosine residues that are phosphorylated by JAKs upon receptor complex activation (Figure 1). TYK2 also associates with the gp130 receptor chain, yet there is no evidence that gp130-utilizing cytokines rely on TYK2 for signal transduction [5,6]. Note that comprehensive reviews [2,7] provide lists of various other receptors utilizing TYK2-STAT signaling; however, TYK2-STAT activation/utilization is frequently only biochemically assessed by phosphorylation of critical tyrosine residues and cannot be put on a level with dissected downstream cellular activities. Here we review the cytokines which clearly transduce the TYK2 phosphorylation events into downstream physiological changes (Figure 1).

The biological relevance for TYK2-dependent cytokines activating STAT3 is best established for the IL-10R2 utilizing IL-22 [8,9] and the IL-12Rβ1-utilizing IL-12 and IL-23 [10,11,12]. IL-22 is a central cytokine in tissue-barrier function, wound healing, and epithelial homeostasis and repair. Cancer promoting, as well as restraining, functions were described [13,14]. IL-23 is a key mediator of inflammation, bridges innate and adaptive immune responses, and is known to support tumorigenesis and metastasis [15,16]. IL-12 is central in promoting cell-mediated immunity to infection and cancer [12]. However, this anti-carcinogenic function can be counteracted by IL-12-STAT3-promoted production of pro-carcinogenic IL-23 [17]. While STAT3 is activated by type I and III IFN stimulation in various cell types, its biological functions in the IFN responses are less clear. Growing evidence suggests that STAT3 is a negative regulator of type I IFN activities, thereby providing a pro-viral and pro-survival cellular program [18]; there is, however, also a report on an opposite, i.e., anti-viral activity of STAT3 [19]. The role of TYK2 in IL-10 signaling through STAT3 is not entirely clear and may be cell type- or context-dependent [6]. The double-edged role of IL-10 in immunity and cancer is reviewed elsewhere [9,20]. IL-19, IL-20, IL-24, and IL-26 (absent in mice) constitute a subfamily within the IL-10 cytokine family and signal primarily through activation of STAT3 [9]. Activation of TYK2 at the respective receptors has not been formally shown but can be inferred from the receptor-chain composition. As this subfamily constitutes relatively recently discovered cytokines, cellular responses are still poorly defined, and we refer to recent publications and reviews for a potential cancer connection [21,22,23]. Lastly, without specification of the cytokines involved, TYK2 via STAT3 was reported to be crucial for the mediation of cell death in an auto-inflammatory context [24].

STAT5, in contrast, is not among the primarily activated STATs downstream of TYK2 (Figure 1) and occurs dependently on cell type and differentiation stage, in response to type I and III IFNs [25,26], IL-10R2-, and IL-12Rβ1-receptor family cytokines [9,12]. Neither a cytokine-TYK2-STAT5 axis nor its significance have been established under physiological conditions.

## 2. Aberrant Expression and/or Activity of TYK2 in Cancers

The JAK-STAT pathway is recognized as a core cancer pathway [27] and directly contributes to all hallmarks of cancer [28]. Oncogenic JAK activity can originate from aberrant JAK expression, deregulated upstream signals, GOF mutations, or generation of fusion proteins, as well as loss of negative feedback regulation [2,29,30,31]. Initially, cancer research focused on JAK1-3, while the TYK2 impact on disease was predominantly studied in inflammatory and (auto-)immune diseases [32,33]. Table 1 summarizes the literature on constitutive or hyperactivated TYK2, as well as GOF-mutated TYK2 and the resulting activation of STATs in cancers.

### 2.1. Aberrant TYK2 Levels

In vitro studies with overexpressed JAKs revealed that aberrant TYK2 levels lead to cellular transformation with constitutive phosphorylation of STAT3 [34]. An unusually high expression of TYK2 associated with or causative for carcinogenesis (reviewed [35]) was described for various cancer cell lines and samples from patients suffering from prostate [36,37], ovarian [38], cervical [39], and breast cancer [40,41], as well as malignant peripheral nerve sheath tumors (MPNST) [42,43]. Conflictingly, lowered TYK2 levels in tumor samples and sections (tumor cells and stroma) are generally considered to be an unfavorable prognostic marker (e.g., [44], www.proteinatlas.org). This is supported by a recently published meta-analysis of JAKs and STATs in hepatocellular carcinoma (HCC) patients, where normal or higher TYK2 levels correlated with longer survival and were found in healthy tissue [45]. The underlying reason for these conflicting reports may be attributed to the anti-proliferative/pro-apoptotic and/or tumor surveillance properties of TYK2 [5], as well as the undetermined tumor cell intrinsic and extrinsic state of TYK2. The important role of TYK2 in immune-surveillance is also in line with findings in patients who carry mutated TYK2 alleles which lead to loss of TYK2, lowered TYK2 levels [46], or expression of kinase-inactive TYK2 [47,48,49], and that show primarily immunodeficiencies. Nonetheless, proteomics suggested that low TYK2 facilitates local metastasis in breast cancer [50], and a comprehensive screen for protein tyrosine kinase variants in numerous cancer cell lines identified splice variants that render TYK2 inactive [51]. On a molecular mechanistic level, the cell intrinsic tumor-promoting consequences of low TYK2 or LOF of TYK2 currently remain elusive.

### 2.2. Aberrant Activation of TYK2

A comprehensive list of receptors (over-)expressed in various cancer types which allows us to deduce putative upstream signals involved in hyperactivation of TYK2 was compiled recently [7]. Primary hematological neoplasm (ALCL, anaplastic large cell lymphoma; T-ALL, T cell acute lymphoblastic leukemia) patient samples and cell lines were shown to be dependent on TYK2 activated by upstream IL-10 and/or IL-22 signals and established an upregulation of anti-apoptotic BCL2 family members via STAT1 and/or STAT3 [52,53]. A similar high TYK2-STAT1/3-BCL2 axis was found in MPNST [43]. Cytotoxic T-lymphocyte-associated antigen 4 (CTLA4, CD152) is mainly expressed on T cells and is a well-established immune checkpoint. CTLA4 signaling is initiated through binding to CD80 (B7-1) or CD86 (B7-2) on the surface of antigen-presenting cells (APCs). Ectopic expression of CTLA4 was found on diverse B-cell lymphoma. Mechanistically, it was established that CD86-CTLA4 engagement resulted in recruitment/activation of TYK2, which, in turn, led to a STAT3-driven tumor-promoting transcriptional program [54]. A STAT-independent involvement of activated TYK2 in fibroblast growth factor 2 (FGF-2) mediated escape from drug-induced death was reported for a sarcoma cell line [55].

### 2.3. TYK2 Mutations

Oncogenic JAK2 with the prominent JAK2^V617F^ mutation found in over 50% of myeloproliferative neoplasia (MPN) patients [56] is the paradigm for the understanding of structure/function relations of JAK activity [57,58,59] and for the general alertness of the cancer field for mutated JAK family members as potential oncogenes. TYK2 joined the club of GOF-mutated JAKs causative for patient hematopoietic malignancies only recently: In 2013, the first TYK2 GOF point mutations were found in T-ALL cell lines and characterized to have transforming capacity via STAT1 and a BCL2 family member [53]. With respect to biochemical studies, the first GOF mutation of TYK2 was V678F, which is the homologous mutation to JAK2^V617F^ [60,61]. Until now, this mutation was not found in patients. The only mutation reported in a public cancer genome database (www.stjude.cloud) for this residue is the V678L mutation, albeit with unknown structure/function consequences. Point mutations at the TYK2 locus are distributed throughout the whole gene body, with GOF mutations—similar to the other JAKs—primarily accumulating in the JH1 and JH2 domains ([2,5] and see public databases, e.g., Genomic Data Commons of the National Cancer Institute [62,63], Catalogue of Somatic Mutations in Cancer (COSMIC [64], and cBioPortal for Cancer Genomics [65,66]).

In addition to the somatic cancer cell mutations, two GOF TYK2 germline mutations (P760L and G761V) were found in pediatric patients developing several de novo leukemias. These mutations are located in the JH2 pseudokinase domain of TYK2 and are predicted to attenuate the negative regulation on the JH1 kinase domain, leading to constitutively activated TYK2 [67].

A prominent germline TYK2 mutation is P1104A/V, which was first found to be associated with solid and hematopoietic cancers [68,69] and later with immunological and inflammatory disorders (reviewed in [5]). While analyzing MPNST tumor samples, it was proposed that TYK2^P1104A^ is an unfavorable prognostic marker for the disease [42]. Notably, this study solely genotyped the somatic cancer cells and overlooked that this mutation impairs TYK2 catalytic activity; cellular signaling, however, is not completely abrogated, and the detected induction of BCL2 expression might favor an anti-apoptotic program [69,70]. Recent studies show that TYK2^P1104A^ is a LOF mutation, because patients homozygous for this allele are either susceptible to microbial infection or protected from autoimmune disease [47,49,71]. These mechanistic and phenotypic features of TYK2^P1104A^ were confirmed in independent mouse models [48,71].

### 2.4. TYK2 Fusion Proteins

Chromosomal rearrangements account for a number of driver kinase fusion genes in cancer [72,73,74]. The first fusion kinase involving a JAK was TEL-JAK2, consisting of a 3′ portion of JAK2 and a 5′ region of TEL, a member of the ETS transcription factor family [75]. This chromosomal translocation is found in T-ALL in patients [75] and transgenic mice expressing TEL-JAK2 develop T-cell leukemia [76]. In vitro studies with a TEL-TYK2 fusion showed constitutive activation of STAT1/3/5 and transforming capacities [77], albeit respective translocations have not yet been identified in patients. As observed for GOF-mutated JAKs, JAK2 kinase fusions occur most frequently compared to the other JAKs, which suggests that the JAK2 locus is a mutation and rearrangement hotspot [56,78,79]. The first leukemia patients carrying TYK2 fusion genes described were combinations of the TYK2 kinase domain and a part of the pseudokinase domain with 5′ portions of nucleophosmin (NPM) 1, polyadenylate binding protein (PABPC) 4, or the transcription factors MYB or NFκB2 [80,81,82]. Structurally and mechanistically, the TYK2 fusion proteins lack the negatively regulating function of the pseudokinase (JH2) domain leading to a GOF kinase activity and hyperactivity of STAT3 and depending on the cellularity also STAT1 and 5 (reviewed in [5,58,59]).

Subsequent analysis of patient samples and cell lines [83,84,85,86,87] and screening of cancer data sets revealed more than 50 chromosomal *TYK2* rearrangements found mostly in hematological, but also in solid cancers [88]. For the fusions, it is currently not known if they contribute as driver oncogenes to early tumorigenesis or are rather the result of genomic instability at later tumor stages [89]. Recently, chromothripsis was identified as a new type of chromosomal rearrangement during carcinogenesis. Based on a single chromosome-shattering event and DNA repair complex, intra- and interchromosomal rearrangements, such as fusion genes, are produced within a few cell cycles. If the fusion event(s) allow for growth or survival advantages, a cancer driver gene might be generated [90,91]. Chromothripsis was assigned to genomic alterations in childhood cancer [92], and mechanistically it is caused by defects in the nuclear envelope composition or formation and failures during mitosis [93]. It is tempting to speculate that the remarkably high number of described TYK2 fusions were—at least in part—generated through chromothripsis and thus might act as driver mutations.

## 3. Tumor-Promoting Activities of (Hyper-)Active TYK2

The molecular contribution of TYK2 signaling and known protein–protein interactions to the hallmarks of cancer were reviewed previously [5,28]. Here, we highlight the latest findings on the consequences of TYK2 hyperactivity in cancer cells.

### 3.1. TYK2 Activation of (Oncogenic) STAT Signaling

As shown in Figure 1, the heterodimeric cytokine receptors with engagement of TYK2 are capable of activating all STATs. Hyperactive, GOF-mutated TYK2 or TYK2 fusions in oncogenic settings preferentially lead to aberrant activation of STAT1, STAT3, and STAT5. The oncogenic potential of STAT3 and STAT5 was recognized early on and is well documented [94,95]. STAT1 was initially considered to exert tumor suppressor functions, and its oncogenic potential emerged more recently [96,97,98].

STAT1/3/5 were found hyperactivated in patient-tailored cell lines with activated TYK2 [53], as well as carrying somatic or germline TYK2 GOF mutations [53,67] or TYK2-NPM1 and -NFkB2 fusions [80,82]. In other tumor samples or experimental tissue culture settings, STAT3 only, or other dual combinations of activated STAT1/3/5, are described (see Table 1).

Interestingly, TYK2 does not only phosphorylate the major phosphorylation site Y705 in STAT3, but also Y640, which represses STAT3 activation [99]. This phosphorylation site in STAT3 is often mutated in cancers [100,101]. Neither the general (patho-)physiological impact nor the contribution to malignancies of this phosphorylation event are currently known.

### 3.2. TYK2 Stimulation of Tumor Cell Invasion

The families of tight junction proteins claudins (CLDNs) and of matrix metalloproteinases (MMPs) are central for the invasion of tumor cells and, in consequence, metastasis formation [106,107]. Recent studies show that, in liver and lung carcinoma, high levels of CLDN9/12/17 caused activation of TYK2 and STAT1/3 and promoted metastasis [102,104,105]. The promoters of various MMP genes harbor STAT binding sites, and many MMPs are transcriptionally activated through TYK2-associated cytokine receptors [108,109]. Gene-targeted mice revealed that TYK2 and STAT1 are required for expression of MMP2/9/14 under inflammatory conditions [110]. Biochemical studies showed that, dependent on context and inflammatory conditions, MMP1/3 induction involves STAT1 alone [108] or also STAT3 [111]. In a hematopoietic tumor TYK2-STAT3 induced MMP9 and tumor cell invasiveness [54] and in a solid tumor TYK2-STAT3 signaling induced MMP1 expression [103].

The urokinase-type plasminogen activator (uPA)/receptor (uPAR) system is central for a cascade of proteolytic events, including activation of MMPs, which allow for tumor cell migration and metastasis [112]. Signaling via uPAR involves TYK2 and PI3K [113], and, at the post-transcriptional level, TYK2 inhibits the accumulation of plasminogen activator inhibitor (PAI) 2 [114]. In prostate cancer, high levels of TYK2 correlate with invasion and metastasis [36,37]. In an ovarian cancer cell line pY-STAT3 co-localizes with TYK2 and JAK2 at focal adhesions, and hyperactive STAT3 was shown to promote cancer cell motility [38]. Without providing molecular details, a mouse model for aggressive lymphoma showed reduced tumor cell invasiveness upon loss of TYK2 [115]. In addition, without providing molecular insights, a siRNA screen assessing the role of the tyrosine kinome in metastasis formation identified TYK2 as a promoter of invadopodia, which are cellular structures characteristic for tumor cell migration [116,117]. Connexin43 (Cx43) is the most widely expressed member of a large family of transmembrane proteins involved in gap junction formation. Cx43 can be both pro- and anti-tumorigenic, e.g., by promoting invasion and metastasis and by acting as a tumor suppressor [118,119]. TYK2 was found to play a dual role in regulation of Cx43: On the one hand, TYK2 is capable of directly phosphorylating Cx43, thereby decreasing its stability; on the other hand, angiotensin II-activated TYK2 increased Cx43 levels in a STAT3-dependent manner [120]. This regulatory loop has not yet been studied in the context of carcinogenesis. Furthermore, knockdown of TYK2 reduced migration of breast cancer cell lines [50].

### 3.3. TYK2 Prevention of Apoptosis

IFNs in general are capable of promoting apoptosis of cancer cells [121]; hence, provided that IFN stimulus and responsiveness in the tumor is given, TYK2 acts tumor suppressive. Tumor cells are able to resist cell death by upregulation of anti-apoptotic BCL-2 family members [122,123]. TYK2 was shown to drive either in a STAT1- and/or a STAT3-dependent manner or in a STAT-independent but ERK1/2-dependent manner high expression of BCL-2 [43,53,55] or its family members BCL-2L1 [54] and MCL1 [52,55]. In contrast, an in vitro study demonstrated that TYK2 physically interacts with SIVA-1 and promotes SIVA-1 mediated apoptosis, as well as inhibits BCL-2 [124].

### 3.4. TYK2 Crosstalk to Oncogenes and Proto-Oncogenic Pathways

In a mouse model of ALCL, as well as in patient cells, TYK2 showed co-operativity with the oncogenic fusion kinase NPM-ALK [52]. In contrast, no co-operation of TYK2 with mutated FLT3-ITD or JAK2^V617F^ in MPN mouse models was found [125,126]. The latter is consistent with the observation that, in JAK2^V617F^ MPN patients (see below) resistant to pharmacological JAK2 inhibition, only JAK1, and not TYK2, leads to heterodimeric STAT activation, despite both kinases show equal tyrosine phosphorylation at the activating loop [127]. This is to be expected, since, in contrast to the other JAKs, loss of TYK2 at heterodimeric JAK-associated cytokine receptors leads only to a partial impairment in signaling [5,6], and, as experimentally described for the IFNAR receptor, TYK2 is the subordinated JAK at cytokine receptors [128,129].

Early biochemical studies suggest that, upon type I IFN treatment, TYK2 interacts with various proto-oncogenes, including the guanine nucleotide exchange factor 1 VAV, the E3 ubiquitin-protein ligase C-CBL, and the SRC family tyrosine kinases FYN and LYN [130,131,132,133,134]. The importance of these PPIs for tumorigenesis is currently unknown. In cancer samples or cell lines, TYK2 was found to cooperate with other oncogenic effectors and pathways, such as the RAF/ERK [53,55,61], MAPKs [135], PIM1/2 [84], and PI3K/AKT/mTOR pathway [36,53,61]. Reported solely in the context of skin inflammation is the TYK2-STAT3 requirement for expression of IκBζ (encoded by *NFKBIZ*) [136]; however, emerging reports suggest cell-intrinsic oncogenic, as well as tumor-suppressive, functions of IκBζ [137].

The mapped and predicted PPIs of TYK2 based on proteomics [138,139] and next generation sequencing (NGS) are accessible at various open-source databases (for a review, see [140]). The TYK2 kinase domain and a STAT3-based reporter system were used to establish the first mammalian two hybrid kinase substrate sensor (KISS) screening platform [141,142]. These databases and the screening approaches should be systematically exploited to further define and fine tune the TYK2 interactome in health and disease.

## 4. Deactivation and Stabilization of TYK2 under (Patho-)Physiological Conditions

JAK activity is counter regulated by molecule-intrinsic events, such as post-translational modifications (PTMs) and the inhibitory function of the pseudokinase domain [143] as well as by extrinsic inhibitory regulators, such as suppressor of cytokine signaling (SOCS) proteins and protein tyrosine phosphatases (PTPs) [144].

Databases [145,146] provide curated PTMs, but with the exception of the well described activating phosphotyrosines, there is still a lack of information on the properties of JAKs that are defined by PTMs. For TYK2, ubiquitination and phosphorylation are detected at multiple residues and discussed in the context of stability/decay (PhosphoSitePlus^®^, [146]), albeit the (patho-) physiological relevance is unknown.

SOCS proteins are encoded by STAT target genes and are negative feedback inhibitors of JAK signaling. SOCS1 and 3 are the most potent JAK inhibitors because, in addition to recruitment of JAKs to E3 ubiquitination/degradation mediated by all SOCS family members, they also harbor a kinase inhibitory region (KIR), which efficiently shuts down JAK activity by binding to the JH1 domain [147]. Activated JAKs and cytokine receptor chains are dephosphorylated by multiple PTPs [148]. The current literature regarding deactivation of TYK2 by SOCS1/3, the PTPs PTB1B and SHP1, as well as the global impact of SOCS and PTP family members in cancer are reviewed elsewhere [5,149,150,151].

In vitro studies showed that in hematopoietic tumor cells the PTP SHP1 suppresses growth via accelerating the TYK2 protein degradation [152]. In lung cancer cells, overexpression of the E3 ubiquitin ligase seven-in-absentia-2 (SIAH2) accelerates the proteasomal degradation of TYK2, thereby attenuating STAT3 signaling [103].

HSP90 is a chaperone supporting folding, stability, and function of many client proteins, including JAKs and STATs [153,154,155]. Cancer cells frequently use HSP90 to stabilize and/or increase the function of numerous oncogenes, and HSP90 inhibitors have been studied as anticancer drugs for more than two decades [156,157]. Physical interaction of HSP90 with TYK2 was demonstrated in cancer cell lines and confirmed in a proteome-wide assessment of the HSP90 interactome [158,159]. HSP90 inhibitor treatments in various tumor settings showed beneficial effects by reducing the activity of TYK2 or its fusion proteins [158,160,161].

An emerging field is the involvement of noncoding RNAs in the regulation of the JAK-STAT pathway in carcinogenesis [162,163,164]. Recently, the long noncoding RNA (lncRNA) MEG3 in concert with a microRNA (miR-147) was reported to modulate JAK-STAT signaling in chronic myeloid leukemia (CML). Interestingly, the lncRNA was found to physically interact with TYK2, JAK2, and STAT3, thereby diminishing the activity level of STAT3 (and STAT5) [165].

## 5. Pharmaceutical TYK2 Inhibition

The first selective JAK inhibitor (JAKinib) to be tested in humans was tofacitinib, which potently inhibits JAK3 and JAK1, and, to a lesser extent JAK2, and has little effect on TYK2 [166]. Historically, JAKinibs were developed as immunosuppressive drugs for the clinical use in organ transplants and autoimmune diseases [167]. The success story of ruxolitinib, a JAK2 and JAK1 inhibitor which was the first JAKinib approved for treatment of a hematopoietic malignancy, pushed the perception of JAKinibs as anticancer drugs [168,169]. For insight in development and clinical use, as well as side effects of JAKinibs, we refer to the most recent reviews [170,171,172,173].

TYK2inibs are mainly envisaged as therapeutics for treatment of autoimmune and inflammatory diseases [33,174], in which JAKinib selectivity is currently considered not to be of utmost importance [175]. As for the other JAKinibs, the first generation TYK2inibs are directed to the JH1 domain and compete with ATP in binding to the enzymatic pocket. These inhibitors are potent in inhibiting wildtype (overexpressed) TYK2, mutated (hyperactive) TYK2, and TYK2 fusion proteins harboring the JH1 domain. Since the JAKs show high homology in the JH1 domain, it is hard to develop ATP-competing inhibitors with high selectivity for one particular JAK family member [170,172]. A next-generation inhibitor of TYK2 is directed against the JH2 domain and recently passed the phase II clinical trial for psoriasis treatment [176]. A comprehensive report on the high selectivity and the biological effects of this TYK2inib in mouse models, as well as its efficacy in human cells collected from autoimmune patients, was recently published [177]. JH2-specific TYK2inibs are currently further improved, and additional compounds are being developed [178,179,180,181]. The only TYK2inib reported and successfully tested to block TYK2 activity in an oncogenic setting is a JH1-specific TYK2inib [135]. Notably, JH2 domain inhibitors might not be working for treatment of diseases driven by TYK2 fusion genes missing parts of the JH2 domain.

## 6. Conclusions and Future Perspectives

Since the discovery of TYK2 and the JAK-STAT signaling paradigm in the early 1990s, enormous progress has been made in the structural and functional understanding of the linear JAK-STAT axis and the crosstalk of JAKs or STATs to other signaling hubs, as well as the cell type-specific contributions of JAKs and STATs in health and disease. The striking phenotypical similarities between mouse models deficient for TYK2 or engineered to express kinase-inactive TYK2 and human patients carrying the respective germline mutations established TYK2 as a fundamental component in both innate and adaptive immunity. The (patho-)physiological and molecular pathway similarities of TYK2 in human and mice allow for highly informative comparative biomedical studies and efficient translation of basic molecular insights into clinical applications. The use of TYK2inibs in the treatment of immunological and inflammatory diseases is within reach [182] and is also attractive for malignancies with the involvement of hyperactivated TYK2. The role of TYK2 and GOF-mutated TYK2 upstream of oncogenic STAT3—and, less frequently, STAT1—is established, while, up to now, no mechanistic evidence for an oncogenic TYK2-STAT5 axis is given. Mouse models as genetic mimics of kinase-inhibited TYK2 exist [48,71,183,184] and are currently exploited to further dissect the kinase-dependent from the scaffolding functions of TYK2.

In a short-term perspective, work should concentrate on the use of refined TYK2 mouse models that allow studying the kinase-independent and cell type-specific functions, in order to fully in vivo assess TYK2inibs with respect to their benefits and unwanted side effects. Mouse models to study the consequences of aberrant high TYK2 and GOF-mutated TYK2 are underway (K. Wöss, T. Rülicke et al., unpublished). For pharmacological intervention with oncogenic TYK2, TYK2inibs with the highest possible selectivity are required, and efforts should focus on the further development and in vivo testing of these next-generation TYK2inibs.

In a long-term perspective, the further understanding of the TYK2 function requires the in-depth elucidation of the PTMs and the interactome of TYK2 under spatiotemporal conditions. Additionally, computational modelling and structure predictions (e.g., [185]) should complement the attempts to determine the holo-crystal structure of TYK2 and to use high-resolution imaging (e.g., [186]) to gain insight into the structural features of full-length wildtype and mutated TYK2, as well as its conformation bound to various cytokine receptors.

## Figures and Tables

**Figure 1 cancers-11-01728-f001:**
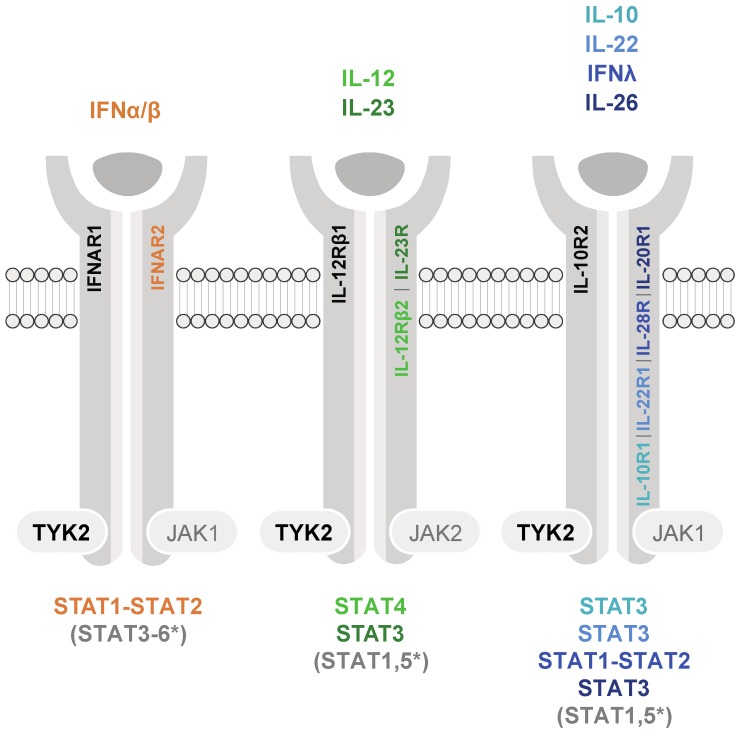
Cytokine receptor families signaling with the participation of TYK2 and JAK1 or JAK2. Cytokines are depicted only upon appearance in humans and mice and proof of TYK2 dependency. The color codes indicate the major STAT(s) activated by the respective cytokines. STAT1-STAT2 heterodimers combine with IFN regulatory factor (IRF) 9 and form the interferon-stimulated gene factor 3 (ISGF3) complex; * STAT activation is dependent on cell type or of less clear biological relevance.

**Table 1 cancers-11-01728-t001:** (Hyper-)active TYK2, GOF-, or LOF-mutated TYK2 and STAT activation in various cancers and cancer cell lines.

TYK2 Status	Disease	Activated STAT	Ref.
Activating somatic mutations (GOF)			
TYK2-G36D; -S47N	T-ALL	STAT1, STAT3	[53] ^(3) (2) (2 *)^
TYK2-731I	T-ALL	STAT1, STAT3, STAT4	[53] ^(3) (2) (2 *)^
TYK2-E957D	T-ALL	STAT1, STAT3, STAT5	[53] ^(3) (2) (2 *)^
TYK2-R1027H	T-ALL	STAT1, STAT3	[53] ^(3) (2) (2 *)^
TYK2-V678F	—	STAT3, STAT5	[61] ^(2 *)^
Inactivating germline mutations (LOF)			
TYK2-P1104A	MPNST	n.d.	[42] ^(4 *) (1)^
TYK2-P1104A	Breast-, colon-, stomach-cancer	n.d.	[68] ^(1)^
TYK2-P1104V	AML	n.d.	[69] ^(5) (1) (2 *)^
Activating germline mutations (GOF)			
TYK2-P760L	B-ALL	STAT1, STAT3, STAT5	[67] ^(3) (1) (2 *)^
TYK2-G761V	T-ALL	STAT1, STAT3, STAT5	[67] ^(3) (1) (2 *)^
Oncogenic fusion proteins (GOF)			
NPM1-TYK2	CD30-positive LPDs	STAT1, STAT3, STAT5	[82] ^(3) (1) (2) (2 *)^
NFkB2-TYK2	ALCL	STAT1, STAT3, STAT5	[80] ^(3) (1) (2 *)^
ELAVL1-TYK2	AML	STAT3, STAT5	[84] ^(2)^
PABPC4-TYK2	ALCL	n.d.	[80] ^(1)^
TEL-TYK2	—	STAT1, STAT3, STAT5	[77] ^(2 *)^
MYB-TYK2	Ph-like ALL	n.d.	[81] ^(1)^
High wildtype TYK2 levels			
TYK2 WT	T-ALL	STAT1, STAT3, STAT4, STAT5	[53] ^(1) (2) (2 *)^
TYK2 WT	ALCL	STAT1, STAT3	[52] ^(4) (1) (2)^
TYK2 WT	Hepatocarcinoma	STAT1, STAT3	[102] ^(3 *) (2 **)^
TYK2 WT	MPNST	STAT1, STAT3	[43] ^(4) (?) (1) (2)^
TYK2 WT	B-cell lymphoma	STAT3	[54] ^(3 *) (2)^
TYK2 WT	Lung cancer	STAT3	[103] ^(3 *) (1) (2) (2 *)^
TYK2 WT	Hepatocarcinoma	STAT3	[104] ^(3 *) (2 **)^
TYK2 WT	Ovarian cancer	STAT3	[38] ^(3 *) (2)^
TYK2 WT	Prostate cancer	n.d.	[36] ^(4) (1) (2)^
TYK2 WT	Prostate cancer	n.d.	[37] ^(4) (1) (2)^
TYK2 WT	Osteosarcoma	no	[55] ^(3 *) (2)^
TYK2 WT	Breast cancer	n.d.	[40,41] ^(4) (1) (2)^
TYK2 WT	Squamous cervical carcinoma	n.d.	[39] ^(4) (1)^
TYK2 WT	MPNST	n.d.	[42] ^(4) (?) (1)^
TYK2 WT	Lung cancer	STAT1	[105] ^(2 **)^
Low wildtype TYK2 levels			
TYK2 WT	Breast cancer (metastatic)	n.d.	[50] ^(6) (1) (2)^

— Unrelated to disease, in vitro findings in stable cell lines, ^(1)^ found in patient samples and primary material, ^(2)^ in vitro findings endogenous TYK2 expression, ^(2 *)^ in vitro findings exogenous TYK2 expression, ^(2 **)^ in vitro findings exogenous claudin expression, ^(3)^ phosphorylated mutated TYK2 protein, ^(3 *)^ phosphorylated wildtype TYK2 protein, ^(4)^ high levels of wildtype TYK2, ^(4 *)^ high levels of mutated TYK2, ^(5)^ reduced levels of phosphorylated mutated TYK2, ^(6)^ reduced levels of wildtype TYK2, and ^(?)^ not specified if wildtype or mutated TYK2. Please note that some references did not study the activation of all STATs and that not all described STATs in the table are active in all cell systems used. Ph = Philadelphia, and n.d. not determined.

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
