# Peer review of "TYK2: An Upstream Kinase of STATs in Cancer"

_cancers, 2019, doi:10.3390/cancers11111728_

Round 1
Reviewer 1 Report
This is an intereseting and extensive overview of TYK2 functioning in various cancers types. The review addresses its role in STAT signalling, the mechanisms of expression deregulation, including mutations, gene fusions and overexpression. Several mechanisms of action are discussed. Perhaps, in the final concluding paragraph the authors can emphasize more explicitely the limitations of our knowledge to give more direction to ongoing work and, perhaps, aspects that desrve more attention in the field.
I have only some minor additional comments:
Page 4, lines 149-150
With respect to the phrase: “This chromosomal translocation leads to T-ALL in patients”, whereas it is likely driving the T-ALL, it is not enough to initiate leukemia (as non of the driving fusions is). Please avoid the formulation “leads to”
Page 7,lines 129-250
One word is missing in the sentence “however, emerging reports suggest cell-intrinsic oncogenic as well as tumour suppressive functions of….. [133].” (TYK2)
Page 7, lines 252-254
Please change into: “The TYK2 kinase domain and a STAT3 based reporter system have been used to establish the first mammalian two hybrid kinase substrate sensor (KISS) screening platform”
Author Response
Detailed information on changes made in manuscript: Cancers-618683 “TYK2 – an upstream kinase of STATs in cancer“:
Comments of the Reviewer 1:
This is an intereseting and extensive overview of TYK2 functioning in various cancers types. The review addresses its role in STAT signalling, the mechanisms of expression deregulation, including mutations, gene fusions and overexpression. Several mechanisms of action are discussed. Perhaps, in the final concluding paragraph the authors can emphasize more explicitely the limitations of our knowledge to give more direction to ongoing work and, perhaps, aspects that desrve more attention in the field.
I have only some minor additional comments:
Point 1: Page 4, lines 149-150
With respect to the phrase: “This chromosomal translocation leads to T-ALL in patients”, whereas it is likely driving the T-ALL, it is not enough to initiate leukemia (as non of the driving fusions is). Please avoid the formulation “leads to”
Previous page 4 lines 149-150; now page 4 lines 153-154
We changed the text according to the suggestions of the reviewer.
Point 2: Page 7,lines 129-250
One word is missing in the sentence “however, emerging reports suggest cell-intrinsic oncogenic as well as tumour suppressive functions of….. [133].” (TYK2)
Previous Page 7 lines 129 (249)-250; now page 7 lines 253-254
We formatted the text accordingly.
Point 3: Page 7, lines 252-254
Please change into: “The TYK2 kinase domain and a STAT3 based reporter system have been used to establish the first mammalian two hybrid kinase substrate sensor (KISS) screening platform”
Previous Page 7 lines 252-254; now page 7 lines 256-258
We changed the text according to the reviewer`s suggestion.
Reviewer 2 Report
Summary
Tyk2 is a member of the JAK kinase family that are traditionally known as key signal mediators in cytokine receptor-mediated signaling pathway. The focal point of this manuscript is to review recent progress of Tyk2 in tumorigenicity, focusing on dysregulation of Tyk2 due to LOF or GOF mutations in human patients with tumors.
Major comments
Although the group has published a review covering the similar topic in January, 2017 (Cytokine 89:209), the manuscript indeed updated the most recent progress of the tumorigenic role of Tyk2. Moreover, the authors also compared the results with different databases, including The Human Protein Atlas, to support their viewpoints and included a section of pharmaceutical Tyk2 inhibitors (Tyki) to give a perspective view for therapeutic potential of Tyki in tumors. These are valuable aspects of the manuscript for readers.
Nevertheless, there are some points that the author may need to clarify further because the concepts seem to be counterintuitive to the oncogenic role of Tyk2.
In line 90-95, the authors pointed out that in vitro studies with overexpressed or hyper-activated Tyk2 is often detected in patients due to its oncogenic properties. However, conflicted results from the Human Protein Atlas database showed that lowered Tyk2 in tumor samples and sections are considered as an unfavorable prognostic marker. The authors need to elaborate the discrepancy between in vitro and in vivo outcome for the tumorigenic role of Tyk2. In line 140-145, the authors stated that in MPNST tumor samples Tyk2P1104A, a LOF mutation, is an unfavorable marker. The authors need to elaborate further for the potential reasons why such a mutation will give a worst outcome for MPNST tumor patients given that Tyk2 is oncogenic.
Minor comments
Line 146 remove bold face of the title “TYK2 fusion proteins” to be consistent with the formatting. Line 249-250 the sentence appears to be truncated and needs to be completed.
Author Response
Comments of Reviewer 2:
Although the group has published a review covering the similar topic in January, 2017 (Cytokine 89:209), the manuscript indeed updated the most recent progress of the tumorigenic role of Tyk2. Moreover, the authors also compared the results with different databases, including The Human Protein Atlas, to support their viewpoints and included a section of pharmaceutical Tyk2 inhibitors (Tyki) to give a perspective view for therapeutic potential of Tyki in tumors. These are valuable aspects of the manuscript for readers.
Nevertheless, there are some points that the author may need to clarify further because the concepts seem to be counterintuitive to the oncogenic role of Tyk2.
In line 90-95, the authors pointed out that in vitro studies with overexpressed or hyper-activated Tyk2 is often detected in patients due to its oncogenic properties. However, conflicted results from the Human Protein Atlas database showed that lowered Tyk2 in tumor samples and sections are considered as an unfavorable prognostic marker. The authors need to elaborate the discrepancy between in vitro and in vivo outcome for the tumorigenic role of Tyk2. In line 140-145, the authors stated that in MPNST tumor samples Tyk2P1104A, a LOF mutation, is an unfavorable marker. The authors need to elaborate further for the potential reasons why such a mutation will give a worst outcome for MPNST tumor patients given that Tyk2 is oncogenic.
Point 1:
Previous page 3 lines 90-95; now page 3 lines 90-97:
We are aware of the discrepency between in vitro and in vivo data existing in the published date. We already discussed that in the respective paragraph. To make this more clear we rewrote this part of the manuscript.
Point 2:
Previous page 4 lines 140-145; now page 4 lines 142-145
The reviewer is right in that it might seem counterintuitive that a LOF mutation of TYK2 could give worse outcome to MPNST tumor patients if TYK2 is discussed as an oncogene. We rephrased this part.
Point 3:
Line 146 remove bold face of the title “TYK2 fusion proteins” to be consistent with the formatting. Line 249-250 the sentence appears to be truncated and needs to be completed.
Previous line 146 now line 150 :
Formatting changed accordingly
Previous lines 249-250 now lines 253-254:
Sentence completed and formatting changed accordingly.